# The Systemic Effect of Ischemia Training and Its Impact on Bone Marrow-Derived Monocytes

**DOI:** 10.3390/cells13191602

**Published:** 2024-09-24

**Authors:** Gustavo Falero-Diaz, Catarina de A. Barboza, Katherine Kaiser, Keri A. Tallman, Christopher Montoya, Shailendra B. Patel, Joshua D. Hutcheson, Roberta M. Lassance-Soares

**Affiliations:** 1DeWitt Daughtry Family Department of Surgery, Leonard M. Miller School of Medicine, University of Miami, Miami, FL 33136, USA; gxf309@med.miami.edu (G.F.-D.); cxm2450@med.miami.edu (C.M.); 2Department of Medicine, Leonard M. Miller School of Medicine, University of Miami, Miami, FL 33136, USA; catarina.barboza@miami.edu; 3Department of Biomedical Engineering, Florida International University, Miami, FL 33174, USA; kkais004@fiu.edu (K.K.); jhutches@fiu.edu (J.D.H.); 4Department of Chemistry, Vanderbilt University, Nashville, TN 37235, USA; keri.a.tallman@vanderbilt.edu; 5Department of Internal Medicine, Division of Endocrinology, Diabetes, and Metabolism, University of Cincinnati, Cincinnati, OH 45221, USA; sbpatel@ucmail.uc.edu

**Keywords:** monocytes, inflammation, systemic effect, ischemia training, extracellular vesicles, hindlimb ischemia

## Abstract

Objective: Monocytes are innate immune cells that play a central role in inflammation, an essential component during neovascularization. Our recent publication demonstrated that ischemia training by 24 h unilateral occlusion of the femoral artery (FA) can modify bone marrow-derived monocytes (BM-Mono), allowing them to improve collateral remodeling in a mouse model of hindlimb ischemia. Here, we expand on our previous findings, investigating a potential systemic effect of ischemia training and how this training can impact BM-Mono. Methods and Results: BM-Mono from mice exposed to ischemia training (24 h) or Sham (same surgical procedure without femoral artery occlusion–ischemia training) procedures were used as donors in adoptive transfer experiments where recipients were subjected to hindlimb ischemia. Donor cells were divided corresponding to the limb from which they were isolated (left—limb previously subjected to 24 h ischemia and right—contralateral limb). Recipients who received 24 h ischemic-trained monocytes isolated from either limb had remarkable blood flow recovery compared to recipients with Sham monocytes (monocytes isolated from Sham group—no ischemia training). Since these data suggested a systemic effect of ischemic training, circulating extracellular vesicles (EVs) were investigated as potential players. EVs were isolated from both groups, 24 h-trained and Sham, and the former showed increased expression of histone deacetylase 1 (*HDAC1*), which is known to downregulate 24-dehydrocholesterol reductase (*Dhcr24)* gene expression. Since we previously revealed that ischemia training downregulates *Dhcr24* in BM-Mono, we incubated EVs from 24 h-trained and Sham groups with wild-type (WT) BM-Mono and demonstrated that WT BM-Mono incubated with 24 h-trained EVs had lower gene expression of *Dhcr24* and an HDAC1 inhibitor blunted this effect. Next, we repeated the adoptive transfer experiment using Dhcr24 KO mice as donors of BM-Mono for WT mice subjected to hindlimb ischemia. Recipients who received Dhcr24 KO BM-Mono had greater limb perfusion than those who received WT BM-Mono. Further, we focused on the 24 h-trained monocytes (which previously showed downregulation of *Dhcr24* gene expression and higher desmosterol) to test the expression of a few genes downstream of the desmosterol pathway, confirm the Dhcr24 protein level and assess its differentiation in M2-like macrophage phenotype. We found that 24 h-trained BM-Mono had greater expression of key genes in the desmosterol pathway, such as liver X receptors (*LXRs*) and ATP-binding cassette transporter (*ABCA1*), and we confirmed low protein expression of Dhcr24. Further, we demonstrated that ischemic-trained BM-Mono polarized towards an anti-inflammatory M2 macrophage phenotype. Finally, we demonstrated that 24 h-trained monocytes adhere less to endothelial cells, and the same pattern was shown by WT BM-Mono treated with Dhcr24 inhibitor. Conclusions: Ischemia training leads to a systemic effect that, at least in part, involves circulating EVs and potential epigenetic modification in BM-Mono. These ischemic-trained BM-Mono demonstrated an anti-inflammatory phenotype towards M2 macrophage differentiation and less ability to adhere to endothelial cells, which is associated with the downregulation of *Dhcr24* in those cells. These data together suggest that Dhcr24 might be an important target within monocytes to improve the outcomes of hindlimb ischemia.

## 1. Introduction

Monocytes, as well as macrophages, are part of the innate immune system, playing an important role in the neovascularization process during peripheral arterial disease (PAD) and critical limb ischemia (CLI) scenarios [1]. Approximately 2 million people in the US are reported to suffer from CLI. This condition is known to have the highest mortality rates of all atherothrombotic diseases, with rates as high as 20% at 6 months and up to 50% at 5 years [2]. Although it is difficult to predict the economic burden of CLI because most studies only include hospitalized patients, a recent US study estimated the annual Medicare costs of CLI at USD 12 billion [3]. Thus, there is an urgent unmet medical and societal demand to promote limb neovascularization of these patients. Arteriogenesis is the remodeling of pre-existing collateral vessels that works as a natural bypass. This process, along with the hypoxia-mediated sprouting of new capillaries (angiogenesis), ensures that adequate blood reaches the ischemic area [4,5]. Inflammation plays a major role during neovascularization. Polarization of macrophages to an anti-inflammatory phenotype known as M2-like macrophages is important during neovascularization [6,7,8,9]. Anti-inflammatory monocytes lead to proper arteriogenesis and angiogenesis by differentiating into M2 macrophages and secreting anti-inflammatory factors [10,11].

Previous pre-clinical reports have shown that adoptive transfer of bone marrow-derived or circulating monocytes in recipient mice subjected to hindlimb ischemia significantly improves perfusion [12,13]. Our recent study showed that a single insult of ischemia performed by unilateral femoral artery ligation, a process called ischemia training, leads to BM-Mono reprogramming, which had a beneficial impact in a mouse model of limb ischemia [1]. The adoptive transfer of the reprogrammed BM-Mono improved perfusion and vascular remodeling in recipient mice after hindlimb ischemia [1] and these monocytes presented a particular signature, with downregulation of 24-dehydrocholesterol reductase (*Dhcr24*) and accumulation of desmosterol. Desmosterol is a lipid and an intermediate product in cholesterol synthesis recognized as an immune modulator [14] that can interfere with macrophage polarization [15,16]. The “Goldilocks principle” of optimal monocyte infiltration for proper neovascularization has shown that excessive inflammation leads to dysfunctional vascular remodeling and leakage. In contrast, insufficient monocytes/macrophages inhibit vascular cell proliferation and matrix degradation, key processes during neovascularization.

The current study aims to extend our previous findings, first by analyzing if ischemia training has a potential systemic effect in monocytes and identifying if this is associated with circulating factors. We then investigate whether these monocytes have anti-inflammatory characteristics. We demonstrate that a unilateral ischemia training insult impacts the BM-Mono from the previous ischemic limb as well as the BM-Mono from the contralateral limb, as both positively respond to permanent hindlimb ischemia by improving perfusion. The data suggest that ischemia training leads to a systemic effect that is in part associated with circulating extracellular vesicles (EVs) containing epigenetic cargos. Moreover, we confirmed, using transgenic knockout mice, that the suppression of *Dhcr24* in BM-Mono is essential for the beneficial response of these cells to ischemic-limb perfusion. We confirmed that ischemic-trained BM-Mono have an activated desmosterol pathway, that they polarize in anti-inflammatory M2-like macrophages, and that they have a lower adhesion rate to endothelial cells.

## 2. Methods

### 2.1. Animals

All animal experiments were approved by the University of Miami Miller School of Medicine Institutional Animal Care and Use Committee (IACUC) and followed NIH guidelines. C57Bl6J male and female mice of an average age of 3 months were obtained from the Jackson Laboratory or bred in our facility and housed in our animal facility.

### 2.2. Ischemia Training

The femoral artery (FA) was exposed from the left leg and occluded distally at the level of the inguinal ligament and proximally at the level of the popliteal bifurcation. In the 24 h-trained group, FA was occluded for 24 h. The next day, the FA was opened. Sham treatment mice underwent the same surgical procedure except for the FA occlusion. The entire process was followed by laser Doppler imaging (LDI) to confirm the procedure’s success. Details of this method can be found in our recent publication [1].

### 2.3. Permanent Hindlimb Ischemia—Recipient Mice

This procedure was performed as described previously [1]. After anesthesia, the mice’s left FA was exposed through a small incision, and a ligature was placed distal to the origin of the lateral caudal femoral and superficial epigastric arteries and below the inguinal ligament. The FA was transected between the sutures. The wound was closed, and postoperative analgesia care was administered [1].

### 2.4. Laser Doppler Imaging—LDI

LDI was used during ischemia training (donor mice) and in the recipient mice subjected to permanent hindlimb ischemia preoperatively, immediately postoperatively, and 3, 7, and 14 days after hindlimb ischemia surgery. Blood flow was measured and quantified in the mice’s feet, and the ratio between ischemic and non-ischemic was determined. Since body temperature could vary between the mice and influence the absolute measurement of the blood flow in the foot, the ratio between the ischemic and non-ischemic foot should always be considered to avoid misinterpretation of absolute values [1].

### 2.5. Monocyte Adoptive Transfer

Monocytes were isolated from C57Bl6J donor mice subjected or not to ischemia training (24 h-trained or Sham) or, in another set of experiments, from C57Bl6 and *Dhcr24* global conditional KO donor mice (bone marrow cells from these KO mice were shipped from Dr. Patel’s lab). Two days after the ischemia training was terminated, donor mice were euthanized and the BM-Mono were isolated. For the adoptive transfer experiment using 24 h-trained and Sham mice, BM-Mono were isolated from the ischemic limb (left leg) or the contralateral limb (right leg). For the *Dhcr24* conditional KO experiment, BM-Mono were isolated from both limbs. Monocyte isolation was performed using a monocyte isolation kit from Miltenyi Biotec B.V. & Co (Bergisch Gladbach, Germany-MACS cat. no. 130-100-629), and the manufacturer’s instructions were followed. Recipient mice were C57Bl6J, subjected to permanent hindlimb ischemia one day before receiving 1 × 10^6^ monocytes via tail vein injection. In this experiment, we used the same delivery method and the number of cells previously used in our recent publication [1].

### 2.6. BM-Mono qRT-PCR

To evaluate if the ischemic-trained BM-Mono showed upregulation of important genes in the desmosterol pathway, these cells were isolated from the Sham and 24 h-trained groups. We also used the same method to evaluate the expression of *Dhcr24* in BM-Mono co-cultured with 24 h-trained and Sham EVs, as mentioned below. After BM-Mono isolation, using the same procedure mentioned above. Total RNA was extracted using an E.Z.N.A Total RNA Kit from Omega Bio-tek (Norcross, GA, USA cat. no. R6834-02), and equal amounts of RNA were reverse-transcribed using a high-capacity cDNA reverse-transcription kit (Applied Biosystems–Waltham, MA, USA). Gene expression of *LXRα*, *LXRβ,* and *ABCA-1* was measured by qRT-PCR, running in 20 μL reactions in a 7300 real-time PCR system machine using TaqMan Gene Expression master mix (Applied Biosystems), according to the manufacturer’s instructions. Data are expressed as relative fold change over the Sham samples. We also used the same cDNA and real-time system to quantify *HDAC1* in ischemic-trained or Sham EVs; however, the RNA from those EVs were isolated by Alpha Nano Tech LLC (Durham, NC, USA).

### 2.7. BM-Mono Western Blot

BM-Mono isolated from Sham and 24 h-trained mice were lysed, and protein concentration was quantified with a DC Protein Assay Kit I from Bio-Rad (Hercules, CA, USA cat 5000111). Twenty-five milligrams of each sample was mixed with a loading buffer, boiled for 5 min, and subjected to SDS-PAGE in 4–20% gel. After electrophoresis, the proteins were transferred to a nitrocellulose membrane, blocked for 60 min with 5% skim milk, and then incubated overnight at 4 °C with primary Dhcr24 polyclonal antibody from Invitrogen (Waltham, MA, USA-1:1000—PA5-76186) or b-actin monoclonal antibody from Sigma (St. Louis, MO, USA-1:5000—A1978). Then, the membrane was washed with TBS-T and incubated with the corresponding secondary antibodies for 60 min at room temperature. After thorough washing of the membrane, the antibodies were visualized with an enhanced chemiluminescent detection system (Advansta WesternBright™ ECL HRP Substrate Kits–Thermo Fisher Scientific, Waltham, MA, USA). Proteins were identified by their predicted molecular mass (Dhcr24, 55 kDa; b-actin, 42 kDa). Protein band intensities from Western blots were quantified by densitometry using ImageJ (version 1.51j8).

### 2.8. Analysis of Extracellular Vesicle Morphology and Content

Plasma EVs were isolated from 24 h-trained and Sham groups and sent to Alpha Nano Tech for RNA isolation/quantification and electron microscopy pictures of the EVs. Mice’s blood was harvested by cardiac puncture and collected in microtubes containing 10 mL of heparin, and the samples were centrifuged for 15 min at 3000 rpm at 4 °C. The supernatants (plasmas) were transferred to new tubes and stored at −80 °C until EV isolation. EVs were purified from plasma samples using qEVoriginal—35 nm Gen 2 Columns (ICO-35) from Izon Science Ltd. according to the manufacturer’s instructions. As mentioned above, the total RNA of EVs was isolated by Alpha Nano Tech to quantify *HDAC1* gene expression by qRT-PCR. For electronic microscopy, copper carbon formvar grids were glow-discharged immediately before loading with the plasma sample for this experiment. Grids were floated on 10 µL of purified plasma EV sample drop for 10 min, washed twice with filtered DI water by floating on a drop of water for 30 s, and negatively stained with 2% uranyl acetate by floating on a drop of stain for 30 s. The grids were blot-dried with Whatman paper and imaged with a Jeol 1230 electron microscope.

For co-culture of EVs and BM-Mono, EVs were isolated from the serum of 24 h-trained and Sham-treated animals via ultracentrifugation at 100,000× *g* for 2 h. WT BM-Mono were co-cultured with EVs from Sham and 24 h-trained groups for 16 h, and the RNA from monocytes was isolated, as detailed in the section above. In another set of experiments, we treated the WT BM-Mono with sodium butyrate at 1 mM for 60 min before adding the EVs and co-cultured them for 16 h. Sodium butyrate is an HDAC1 inhibitor used to assess whether this histone deacetylase contributed to the monocyte reprogramming process. As mentioned in the section above, total RNA extraction and cDNA from these BM-Mono was performed. Gene expression of *Dhcr24* to assess whether EVs are responsible for the downregulation of *Dhcr24* in BM-Mono was measured by qPCR. Data are expressed as relative fold change over the BM-Mono exposed to Sham EVs.

For confocal images, isolated EVs were stained using a PKH lipophilic dye (Millipore Sigma, Rockville, MD, USA-PKH26) following manufacturer instructions and resuspended in PBS. Stained EVs or PBS controls (100 µL) were added to untreated monocyte cultures and incubated for 24 h. Images were obtained using a Nikon C1 confocal microscope (Nikon Instruments Inc., Melville, NY, USA) with a 60× objective.

### 2.9. Differentiation of M2-like Macrophages

BM-Mono isolated from Sham or 24 h-trained mice were plated at 2 × 10^5^ cells per well in 24-well plates in 1 mL of complete medium (RPMI-1640 plus 10% fetal bovine serum inactivated plus penicillin–streptomycin) with m-CSF at 20 ng/mL to induce macrophage differentiation. After 5 days, the medium was removed, and the cells were polarized to M2 using IL4 (30 ng/mL) and IL13 (20 ng/mL) in the presence of m-CSF. After 24 h, the cells were washed 3 times with PBS, lysed, and RNA was extracted, and qRT-PCRs were performed (as mentioned above) for M2 markers (IL-4 and IL-10) [17].

### 2.10. Assay of Monocyte Adhesion to Endothelial Cells

This experiment was performed as per Yang et al. [18] with some modifications. Briefly, C57BL/6 mouse primary aortic endothelial cells (ref. C57-6052) were plated on 24-well plates at 1 × 10^5^ cells/well in 1 mL of complete mouse endothelial cell medium (both from Cell Biologics, Inc. Chicago, IL, USA). The cells were kept in culture for 3 days until they reached confluence. They underwent starvation overnight, and after that, they were activated for 5 h with m-TNFa. Meanwhile, BM-Mono isolated from Sham or 24 h-trained mice were stained with 2 µM of calcein AM fluorescent dye (Invitrogen Waltham, MA, USA) in a free medium at 37 °C for 30 min. Then, they were washed twice with PBS at 300× *g* for 10 min and resuspended in a complete medium, 1 × 10^6^ labeled monocytes were added on top of the endothelial cells, and the plate was incubated at 37 °C, protected from light. After 1 h, the wells were washed three times with PBS. Finally, the fluorescence intensity of the labeled monocytes attached to the endothelial cells was measured in a fluorescent microplate reader at 488 nm of excitation and 517 nm of emission. For the monocyte adhesion assay using WT monocytes, we treated them with a Dhcr24 inhibitor called SH42. We repeated the protocol described above, but before calcein staining, WT BM-Mono were treated with SH42 at 1 µM or DMSO for 16 h and washed 3 × with PBS Desmosterol. quantification was performed as per Tallman et al. Briefly, cell pellets (≈700,000 cells in 100 µL PBS) were lysed with lysis buffer (500 µL: 50 mM HEPES, 150 mM NaCl, 0.1% TritonX100, 0.1% SDS, pH = 7.0 containing protease inhibitors) at 0 °C for 30 min. A 100 µL aliquot was used for sterol analysis and 20 µL aliquot for protein concentration (determined by BCA assay). To each sample, a sterol standard mixture (10 µL), antioxidant mixture (10 µL: BHT (2.5 mg/mL) and PPh_3_ (1 mg/mL) in EtOH), CHCl_3_:MeOH (600 µL, 2:1), and saline (300 µL) were added. The sample was vortexed well and centrifuged (5000 rpm for 1 min) to separate the layers. The CHCl_3_ layer was transferred to a new vial and dried under vacuum on a Thermo Savant SpeedVac (Thermo Fisher Scientific, Waltham, MA, USA). Derivatization and LC-MS/MS analysis of sterols was carried out as described in [19]. Endogenous sterol levels were quantified based on the known internal standard amount and normalized to milligrams of protein.

### 2.11. Data Analysis

Statistical calculations were performed using GraphPad Prism version 9.0.1 (San Diego, CA, USA). All values are expressed as means ± standard error of mean (SEM). The data were tested for normality using either the Shapiro–Wilk test or QQ plots for samples that were too small. Welch’s *t*-test was performed to compare the means of two groups with normally distributed data. We used the nonparametric Mann–Whitney test for data that were not normally distributed. A one-sample *t*-test was performed for statistical comparison to the standard of the qPCR analyzed using the fold-change method. For the flow cytometry data, we used a one-way analysis of variance (ANOVA) with Dunnett’s posttest. Adjusted *p* values < 0.05 were considered statistically significant.

## 3. Results

### 3.1. Adoptive Transfer of Ischemic-Trained BM-Mono Isolated from Contralateral Limb Improves Blood Flow Recovery

Donors’ BM-Mono were successfully isolated from the limb previously in ischemia (left limb) or from the contralateral limb (never ischemic—right limb) and injected into recipient mice 48 h after the ischemia training termination. Recipient mice received 24 h-trained (left or right) or Sham BM-Mono one day after undergoing hindlimb ischemia, and the LDI was performed. There was no difference in blood flow analysis between the groups before ischemia, immediately post-ischemia, and on day 7 after surgery. However, on day 14 after surgery, the recipients who received 24 h-trained BM-Mono from the left or the right limb demonstrated a significant improvement in blood flow recovery (Figure 1). Surprisingly, the BM-Mono from the contralateral limb had the same beneficial effect of improving perfusion in the recipient mice, indicating a potential systemic effect caused by ischemia training.

### 3.2. Ischemic-Trained EVs Have Upregulation of Histone Deacetylase Enzyme (HDAC1) and BM-Mono Reprogramming

Plasma EVs from 24 h-trained and Sham mice had their RNA successfully isolated. *HDAC1* gene expression was significantly higher in EVs from the 24 h-trained group compared to the Sham group (Figure 2A). Moreover, EVs from both groups were co-cultured with WT BM-Mono for 16 h, and the gene expression of *Dhcr24* was quantified in the WT BM-Mono. *Dhcr24* gene expression was downregulated in WT monocytes exposed to ischemic-trained EVs (Figure 2B). This effect was blunted in the presence of sodium butyrate (Figure 2C), an inhibitor of HDAC. In addition, we demonstrated by confocal z-stack imaging that during the co-culture, WT BM-Mono were able to engulf the EVs (Figure 2D). Further, we also showed the presence of EVs in the plasma-purified EV samples from both groups using electron microscopy (Figure 2E). These data together suggest that the upregulation of *HDAC1* in ischemic-trained EVs is responsible for the WT BM-Mono reprogramming with the downregulation of *Dhcr24* gene expression.

### 3.3. Adoptive Transfer of Dhcr24 KO BM-Mono Improves Blood Flow Recovery

Our previous report showed that adoptive transfer using ischemic-trained BM-Mono, which had downregulation of *Dhcr24* and accumulation of desmosterol, had a beneficial effect on limb perfusion and arteriogenesis in recipients subjected to hindlimb ischemia [1]. Further, our current data in this study demonstrated that this beneficial response of ischemic-trained BM-Mono in hindlimb ischemia also occurs using BM-Mono from the contralateral limb, suggesting a systemic effect of ischemia training on monocytes. Here, we confirmed that this beneficial effect of BM-Mono in hindlimb ischemia is caused by *Dhcr24* regulation. We used Dhcr24 KO BM-Mono adoptive transfer in WT recipient mice subjected to hindlimb ischemia. We revealed that recipients receiving Dhcr24 KO BM-Mono had greater limb perfusion than those receiving WT BM-Mono (Figure 3). The Dhcr24 KO BM cells were donated by Dr. Patel from the University of Cincinnati, which allowed us to isolate the BM-Mono.

### 3.4. Ischemia Training Alters the Desmosterol Pathway on BM-Mono and Modifies the Phenotype of Macrophages

To confirm that the desmosterol pathway on ischemic-trained monocytes is activated (since our previous data showed downregulation of *Dhcr24* gene expression and desmosterol accumulation) [1], we confirmed expression of Dhcr24 at the protein level and analyzed gene expression of important players in the downstream pathway of desmosterol. We found significantly lower Dhcr24 protein expression in ischemic-trained BM-Mono (Figure 4A). We also found that *LXRα* and *ABCA1* are significantly upregulated in ischemic-trained monocytes and that there is a strong trend for *LXRβ* upregulation (Figure 4B). Moreover, ischemic-trained BM-Mono gave rise to macrophages like the M2 anti-inflammatory phenotype, since they had upregulation of M2 macrophage markers, *IL-10*, and *IL-4* (Figure 4C). Interestingly, this last finding supports the idea that ischemic-trained BM-Mono are reprogrammed because even after 6 days under culture conditions, these monocytes were able to maintain their anti-inflammatory phenotype compared to Sham monocytes.

### 3.5. Ischemic-Trained Monocytes Have a Lower Adhesion Rate to the Endothelium via Dhcr24 Inhibition

We cultured murine ECs until they reached confluence and added calcein AM fluorescence-labeled ischemic-trained or Sham BM-Mono to allow them to adhere to the ECs for 1 h. Non-adherent BM-Mono were removed by gentle washes before fluorescence measurement. Ischemic-trained BM-Mono had a lower adhesion rate to ECs compared to Sham BM-Mono (Figure 5A). We repeated the same experimental design using WT BM-Mono treated with SH42, a Dhcr24 inhibitor [1,20]. This treatment upregulated desmosterol on these cells and significantly decreased their adhesion rate to ECs compared to DMSO treatment (Figure 5B). These data indicate a reduction in ischemic-trained BM-Mono adhesion to the endothelium, potentially via *Dhcr24* downregulation, as the Dhcr24 inhibitor also decreased WT BM-Mono adhesion.

## 4. Discussion

Our study addresses the systemic effect of ischemia training and its impact on BM-Mono. We expanded on interesting findings from our previous publication [1] and uncovered a potential systemic effect caused by an ischemia insult created by 24 h of unilateral femoral artery occlusion. In this study, we found that (1) ischemic-trained BM-Mono isolated from the contralateral limb also improve blood flow recovery (as BM-Mono from the limb previously in ischemia) in a permanent model of hindlimb ischemia, (2) circulating ischemic-trained EVs influence, at least in part, the reprogramming of the BM-Mono, (3) loss of *Dhcr24* in BM-Mono leads to similar improvement in limb perfusion seen by using ischemic-trained BM-Mono in a mouse model of hindlimb ischemia, and (4) ischemic-trained BM-Mono have an activated desmosterol pathway, an anti-inflammatory phenotype, and a lower endothelial adhesion rate that is potentially associated with Dhcr24 inhibition and higher desmosterol levels.

The contribution of monocytes to improved perfusion in ischemic-limb muscle and their role in generating enough inflammation via endothelium adhesion, transendothelial migration, and differentiation in macrophage has been previously reported [20,21]. However, the amount of inflammation associated with the monocyte adhesion rate and the phenotype of these cells (which can be pro- or anti-inflammatory) is a key factor for a proper neovascularization process [22]. Our recent data have demonstrated that ischemia training can reprogram BM-Mono, changing their lipid profile with downregulation of *Dhcr24* and accumulation of desmosterol, leading to an improvement in limb perfusion and neovascularization in a mouse model of hindlimb ischemia [1]. Dhcr24 is an enzyme that converts lanosterol to 24,25-dihydrolanosterol and desmosterol to cholesterol [16,23,24]. Therefore, when Dhcr24 is low, lanosterol and desmosterol accumulate in the cells [25]. Currently, intermediates of cholesterol pathways, including lanosterol and desmosterol, are recognized as immune modulators [15] with the ability to influence macrophage polarization [24,25]. Accumulation of desmosterol activates LXRs, which are essential for cholesterol homeostasis [14] and have the ability to inhibit pro-inflammatory genes [26], suppress the inflammasome in macrophages [24], and induce the expression of the *ABCA1* transporter [27,28,29,30]. Supporting this data, a knock-in mouse model with overexpression of *Dhcr24*, specifically in myeloid cells, showed downregulation of desmosterol, inactivation of LXR, and exacerbation of atherosclerosis [24]. Our findings have shown that ischemic-trained BM-Mono with low levels of Dhcr24 protein have higher *LXRs* and *ABCA1* gene expression levels, suggesting this pathway’s activation (Figure 4A,B). Additionally, ischemic-trained BM-Mono show reduced endothelial adhesion, which decreases inflammation (Figure 5A), and demonstrate an anti-inflammatory profile achieved by differentiating in M2-like macrophages (Figure 4C), known to be very important during proper arteriogenesis [31] Interestingly, the macrophage polarization protocol also suggested a long-term reprogramming of these ischemic-trained monocytes, since they were able to maintain their anti-inflammatory phenotype even after 6 days under the same differentiation/polarization conditions as the Sham BM-Mono. Further, we found that adhesion of monocytes to the ECs can be regulated by *Dhcr24* expression and consequent desmosterol accumulation, since monocytes treated with Dhcr24 inhibitor and higher desmosterol levels showed decreased capacity to adhere to ECs (Figure 5B). Our data also showed the direct effect of loss of *Dhcr24* in BM-Mono during hindlimb ischemia. The adoptive transfer using Dhcr24 KO BM-Mono as donor cells revealed improvement in perfusion of the ischemic limb of recipient mice (Figure 3). Together, these data support and complement our previous publication, where we showed that ischemic-trained BM-Mono that presented low *Dhcr24* expression improve perfusion and arteriogenesis in a mouse model of hindlimb ischemia [1].

This study highlights that ischemia training reprograms BM-Mono in a positive way by downregulating their *Dhcr24* expression and differentiating them towards less inflammatory M2-like macrophages, which are essential cells to improve outcomes in limb ischemia [1]. Our ischemia training consisted of 24 h of ischemia by femoral artery occlusion followed by 48 h of reperfusion before BM-Mono isolation. Although it is known that reperfusion causes excessive reactive oxygen species, inducing a pro-inflammatory environment and worsening conditions already compromised by reduced blood flow and oxygen [10,32], our data demonstrated a positive reprogramming of the BM-Mono. This positive effect might be explained because we assessed the monocytes inside the bone marrow and not the ones in the ischemic site that were subject to a direct effect of the reperfusion or because those BM-Mono were tested 2 days after the blood flow was restored, when the hostile environment caused by reperfusion was diminished.

Another important point we investigated in the current study is the potential systemic effect of ischemia training in BM-Mono from the contralateral limb and the underlying molecular mechanism that explains this effect. Previous reports have demonstrated an increase in *HIF-1α* and hypoxia areas in the BM of the contralateral limb, followed by unilateral FA occlusion [12]. Recently, we demonstrated a similar response of the contralateral limb to a unilateral FA occlusion. We have shown a similar increase in the expression of “ischemic genes”, *HIF-1α* and *GLUT-1*, as well as in the lipid metabolism genes between BM-Mono derived from the ischemic limb and contralateral limb [1], and we also showed increased circulating *GLUT-1* during limb ischemia. *GLUT-1* is one of the most important targets of *HIF-1α* and is known to be increased under ischemia/hypoxia conditions [1]. To confirm whether a similar gene profile between the BM-Mono from the contralateral limb and previously ischemic limb has any impact on the outcome of hindlimb ischemia, we performed adoptive transfer using BM-Mono isolated from the limb that was previously in ischemia (left) or from the contralateral limb (right). Of note, in our ischemia training protocol, the BM-Mono were isolated 48 h after ischemia termination to ensure that we were not analyzing the acute effect of ischemia. Our data revealed that BM-Mono from the contralateral limb had a similar ability as the BM-Mono from the left limb to improve perfusion in the recipient mice subjected to hindlimb ischemia (Figure 1). To dissect the mechanism underlying the systemic effect of ischemia training, we investigated circulating EVs, which have recently emerged as one of the major forms of intercellular communication [33,34,35] EVs can transport and transfer bioactive molecules, called cargo, such as proteins, metabolites, lipids, and nucleic acid, to recipient cells [34,35]. This cargo changes according to its physiological and metabolic condition [36] and is a major player in recipient cell reprogramming [34]. Previous reports revealed that monocytes accumulate EVs from different cell types that can reprogram their functions [37]. Many histone-modifying epigenetic enzymes participate in gene activation or repression by modifying histones in specific lysine and arginine residues [38,39,40], and *Dhcr24* is one of the genes [41] that has its expression increased by HDAC inhibitors [20,42,43]. Previous studies support our finding that EVs are carriers of HDAC mRNAs [34]. Interestingly, we showed that ischemic-trained EVs have greater levels of HDAC1 transcripts than Sham EVs (Figure 2A). Also, we demonstrated that these plasma ischemic-trained EVs can reprogram WT BM-Mono, decreasing their *Dhcr24* gene expression (Figure 2B), and that HDAC inhibition blunts this reprogramming (Figure 2C).

The limitations of our study include that this was an animal model, which may not fully replicate human pathophysiology, potentially limiting the translation of the findings to clinical settings. Additionally, the sample of this study is small due to the limited number of monocytes that can be extracted from each mouse, which sometimes leads to a pool of few mice for each sample. Lastly, this study focused on a specific gene regulation (*Dhcr24*) in a specific cell type (monocytes), challenging the idea of using a global inhibition of Dhcr24, an important enzyme in the cholesterol pathway.

In summary, we demonstrated that there is a systemic effect caused by ischemia training that reprograms BM-Mono from the contralateral limb in a very similar way to the BM-Mono from the limb previously in ischemia to improve limb perfusion in recipient mice subjected to hindlimb ischemia. This systemic effect is at least in part associated with circulating EVs that increase their *HDAC1* cargo following ischemia training, leading to reprogramming of the BM-Mono by downregulating *Dhcr24* gene expression in recipient cells. Moreover, this study confirmed (by using Dhcr24 KO BM-Mono) that reduction in Dhcr24 in these cells is essential to improving limb perfusion during hindlimb ischemia, and this is potentially the reason that ischemic-trained BM-Mono (which have downregulation of Dhcr24) respond well in the hindlimb ischemia scenario. The reprogramming of BM-Mono by ischemia training with downregulation of Dhcr24 leads these cells to activate the desmosterol pathway, allowing them to differentiate in an anti-inflammatory macrophage phenotype and adhere less to endothelial cells. In conclusion, this study expands our knowledge about the impact of ischemia training in monocytes and confirms an important association of Dhcr24–desmosterol in this process. However, further investigation is needed to dissect the causal effect of Dhcr24 regulation in monocytes during hindlimb ischemia, with particular emphasis on the translation of these findings for PAD/CLI in clinical settings.

## Figures and Tables

**Figure 1 cells-13-01602-f001:**
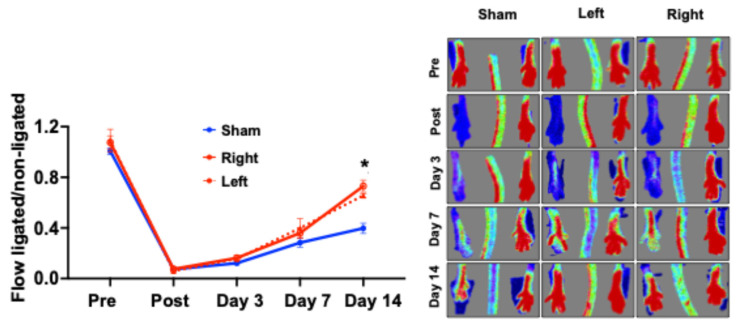
Adoptive transfer of BM-Mono from each limb. Recipient mice subjected to unilateral hindlimb ischemia received Sham BM-Mono or 24 h trained BM-Mono isolated from the left (previously in ischemia) or right limbs. Laser Doppler Imaging showed improved perfusion in recipient mice who received 24 h trained BM-Mono from right or left limbs compared to Sham (n = 8). Error bars represent SEM (* *p* < 0.01 vs. same timepoint).

**Figure 2 cells-13-01602-f002:**
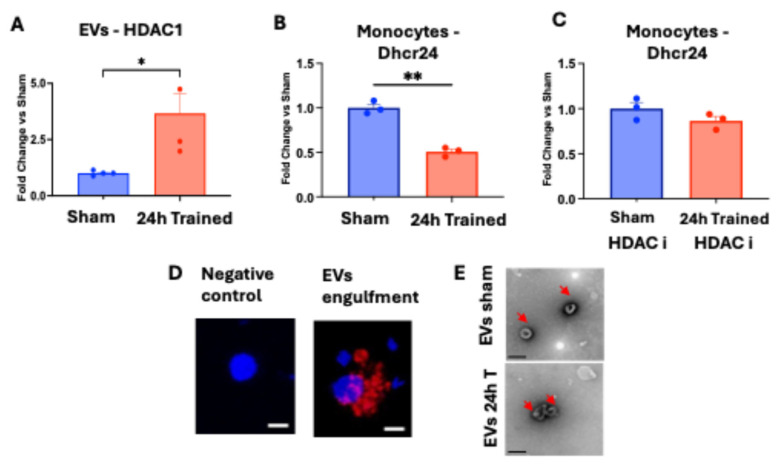
Circulating EVs from Sham and 24 h trained groups were analyzed and co-cultured with wild-type BM-Mono. (**A**) qPCR shows significantly greater *HDAC1* gene expression on ischemic-trained EVs. (**B**) qPCR shows reprogramming of WT BM-Mono with downregulation of *Dhcr24* by 24 h trained EVs. (**C**) qPCR shows that HDAC inhibitor (HDACi) blocks WT BM-Mono reprogramming by rescuing *Dhcr24* expression downregulated by 24 h trained EVs. (**D**) 3D confocal z-stacks images show EVs stained with PKH lipophilic dye (red) engulfed by BM-Mono (Hoechst nuclear stain–blue). (**E**) Representative images of EVs by TEM with negative stain. EVs often appear cup-shaped (shown with red arrows) due to dehydration during sample prep and accumulation of the stain in the formed cavity. Scale bar 200 nm. Error bars represent SEM (* *p* < 0.05, ** *p* < 0.01).

**Figure 3 cells-13-01602-f003:**
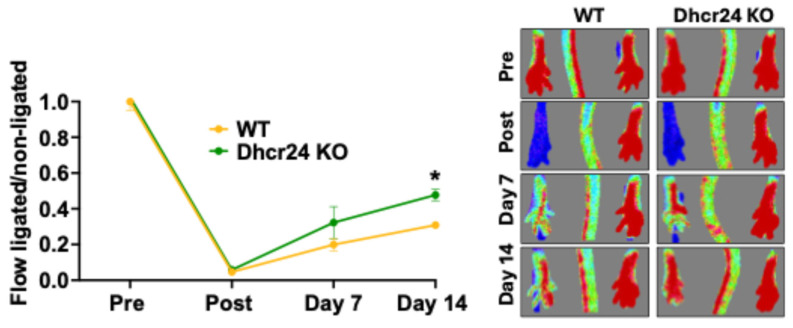
Adoptive transfer using Dhcr24 KO or WT BM-Mono. Wild-type recipient mice subjected to unilateral hindlimb ischemia received WT or Dhcr24 KO BM-Mono. Laser Doppler Imaging demonstrated improved perfusion in recipient mice with Dhcr24 KO BM-Mono (n = 3). Error bars represent SEM (* *p* < 0.05 vs. same timepoint).

**Figure 4 cells-13-01602-f004:**
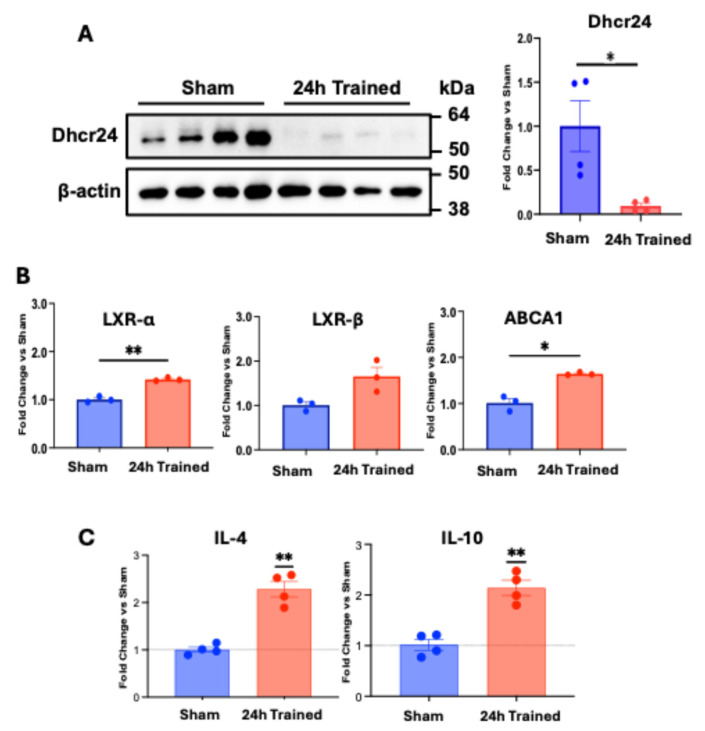
Ischemia training reprogrammed BM-Mono lipid metabolism. BM-Mono were isolated, and protein and gene expression were analyzed. Also, BM-Mono were subjected to macrophage M2 polarization protocol, and gene expression for M2 markers was performed. (**A**) Dhcr24 protein expression was downregulated in ischemic-trained monocytes. (**B**) These cells also showed upregulation of important genes downstream of desmosterol pathway: *LXRs* and *ABCA1*. (**C**) Gene expression of M2 macrophage polarization shows greater levels of M2 markers *IL-10* and *IL-4* on macrophages derived from 24 h trained BM-Mono. “Fold Change vs. Sham” represents the difference of 24 h Trained group related to Sham group expressed in fold change. Error bars represent SEM (* *p* < 0.05, ** *p* < 0.01).

**Figure 5 cells-13-01602-f005:**
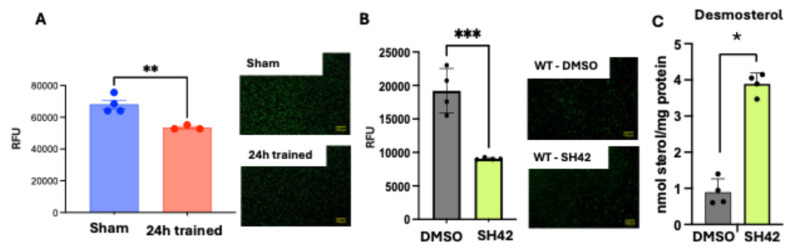
BM-Mono Adhesion Assay. Endothelial cells were plated until they were confluent. Then stained monocytes were added to the culture and allowed to adhere before washes were performed. (**A**) Ischemic-trained BM-Mono showed less adhesion to ECs. (**B**) WT BM-Mono treated with Dhcr24 inhibitor (SH42) also showed less adhesion to ECs and it shows accumulation of desmosterol as expected (**C**). BM-Mono were stained with Calcein AM fluorescent dye, fluorescence was quantified using relative fluorescence unit (RFU) where fluorescence was read at 488 nm excitation and 517 nm emission. Error bars represent SEM (* *p* < 0.05, ** *p* < 0.01, *** *p* < 0.001).

## Data Availability

Data supporting this study can be accessed upon request to the corresponding author.

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
