# Peer review of "The Systemic Effect of Ischemia Training and Its Impact on Bone Marrow-Derived Monocytes"

_cells, 2024, doi:10.3390/cells13191602_

Round 1
Reviewer 1 Report
Comments and Suggestions for Authors
In this study, Falero-Diaz et al. showed that ischemia training leads to a systemic effect leading to epigenetic modification in BM monocytes. They show that this effect was in part due to circulating EVs that carry HDAC1, leading to downregulation of Dhcr24 in BM monocytes. The authors further confirmed this mechanism using Dhcr24-/- mice as donors and showed that the protective effect was blunted. They also showed that these ischemic-trained BM monocytes demonstrated an anti-inflammatory phenotype towards M2 macrophage polarization and that they adhered less strongly to endothelial cells. The data presented in this study is interesting but most of it is generated based on in vitro data supporting their hypothesis. The following comments should be addressed.
1. Although the authors showed that BM monocytes are able to engulf EVs in vitro, this was not shown in vivo. Is there a way for the authors to verify that the BM monocytes are actually engulfing the EVs in vivo?
2. The authors showed that the ischemia trained BM monocytes resulted in a M2 phenotype. They used IL-4 and IL-13 to polarize the macrophages into an M2 phenotype in vitro. However this is not realistic in the in vivo setting, as there would be a lot of other pro-inflammatory factors which would affect the outcome of these macrophages to become M2 macrophages. The authors should culture the macrophages with serum from ischemia mice to determine if the ischemia-trained BM mononocyte-derived macrophages are still able to produce more IL-4 and IL-10.
3. The authors need to revisit their figure legends and citation of the appropriate figures. There are many errors in the manuscript with missing figure annotations. For eg they mention in the results section that monocytes are able to engulf EVs and cite Figure 2C but it should be Figure 2D. The figure legends should also contain more information about how the experiment was performed and not just a one-liner of what it indicates. There is also no consistency in how the figure legends were displayed with some having the alphabet in front while others in the back and it is very confusing to the reader.
Author Response
Comment 1: Although the authors showed that BM monocytes are able to engulf EVs in vitro, this was not shown in vivo. Is there a way for the authors to verify that the BM monocytes are actually engulfing the EVs in vivo?
- Response 1: Thank you. This is a very thoughtful comment, but it was not planned as part of the scope of this manuscript. In our next manuscript, which will be part of our NIH R01 grant just awarded, we will specifically focus on the interaction between the EVs and monocytes and how those ischemic-trained EVs can reprogram the monocytes, showing epigenetic details and in vivo studies. We believe it makes more sense to have a full manuscript focused on how EVs are reprogramming the monocytes.
Comment 2: The authors showed that the ischemia trained BM monocytes resulted in a M2 phenotype. They used IL-4 and IL-13 to polarize the macrophages into an M2 phenotype in vitro. However, this is not realistic in the in vivo setting, as there would be a lot of other pro-inflammatory factors that would affect the outcome of these macrophages’ becoming M2 macrophages. The authors should culture the macrophages with serum from ischemia mice to determine if the ischemia-trained BM monocyte-derived macrophages are still able to produce more IL-4 and IL-10.
- Response 2: Thank you for this comment. We followed the reviewer’s suggestion and tried to perform this experiment to improve our manuscript. However, we tried the experiment twice and couldn’t get RNA from the cells. The cells' viability was low, and we think that might be related to the ischemia serum that was used since we did not have this type of problem before. We believe that at this point, it would not be worth spending months to normalize this experiment if the simple message we want to show here is that ischemia-training monocytes have a higher chance to differentiate in M2-like macrophages when compared to Sham monocytes.
Comment 3: The authors need to revisit their figure legends and citation of the appropriate figures. There are many errors in the manuscript with missing figure annotations. For eg they mention in the results section that monocytes are able to engulf EVs and cite Figure 2C but it should be Figure 2D. The figure legends should also contain more information about how the experiment was performed and not just a one-liner of what it indicates. There is also no consistency in how the figure legends were displayed with some having the alphabet in front while others in the back and it is very confusing to the reader.
- Response 2: Thank you. We revised the entire manuscript, corrected the errors, added the same pattern to the figures, and improved the legends.
Reviewer 2 Report
Comments and Suggestions for Authors
The manuscript “The systemic effect of ischemia training and its impact on bone marrow-derived monocytes” by Falero-Diaz et al is interesting. However, the authors are suggested to address the following comments to improve the quality of the manuscript
1) Abstract Line 26: “compared to recipients with Sham monocytes… a systemic effect of..”. Kindly clarify the meaning of Sham monocytes
2) Line 123~124: quantified in the mice's feet, and the ratio between ischemic and non-ischemic was determined to avoid variation of body temperature between the mice[1]
3) Please clarify the statement “to avoid variation of body temperature between the mice.”
4) Line 131: "24h trained and Smice, BM-Mono … the ischemic limb (left leg) or the". What does Smice refer to?
5) Line134: “Miltenyibiotec MACS (cat. no. 130-100-629), .. instructions “Please correct the manufacturer spelling to Miltenyi Biotec
6) Line 135~136: Recipient mice were C57Bl6J, subjected to permanent hindlimb ischemia one day before receiving 1x106 monocytes via tail vein injection.
Please justify the reason for selecting the tail vein injection as the route of administration of monocytes rather than the ischemic site. Furthermore, how is the dose of 1x106 monocytes optimized? Please show the optimization experiment if the authors have performed any.
7) Line 166: Plasma EVs were isolated from 24h trained and Sham groups and sent to Alpha Nano.
8) Please clarify the process for the isolation of EV or cite the appropriate reference.
9) Kindly include the LDPI scale in Figure 1. It is better to include D1 images to observe the improvement in the mice limb.
10) Please clarify the number of animals taken for the experiment in Fig 2.
11) Line 268: "using electronic microscopy (Figure 2D)...together, suggest that the" Which electronic microscopy was used for this experiment?
12) Fig 2E : Scale bar is missing in EVs by TEM.
113) Fig 3: Please incorporate Day 0 and Day 3 images for better comparison and to observe the progress.
114) Line 356: “and …model of hindlimb ischemia (our ATVB paper)”
The authors are advised to properly reference the paper.
15) Line 299~300: (since our previous data showed downregulation of Dhcr24 gene expression and desmosterol accumulation1) we confirmed protein expression of Dhcr24
Line 324~325: Sham BM-Mono (Figure 5A). We repeated the same experimental design using WT BM- Mono treated with SH42, a Dhcr24 inhibitor1.
What does superscript 1 refer to in the above statements?
16) Is Figure 3: Left panel graph the quantification of LDPI? It is a good idea to incorporate D0 and D3 images as well.
17) Fig 3, n=3 seems the sample size is very small.
Does the result from this sample size give reliable results?
118) Fig 4A: Dhcr expression in both Sham and 24h treated groups, are the 4 wells, protein samples from quadruplets or 4 independent experiments. Please quantify the western blot results.
19) Y axis label is missing in Figure 4B.
Figure 4 C: Kindly clarify label on Y axis “ Fold Change vs Sham is the”
20) Figure 5: Please state the fluorescence unit on the Y axis.
Fig 5a: Kindly clarify which staining is shown in the image.
Please replace the fluorescence image if possible.
21) The authors are suggested to add the Limitation of this study.
222) It is a good idea to include a flowchart to show the experiments/findings to grasp the audience's attention.
Comments on the Quality of English LanguageMinor editing of English language required
Author Response
Comment 1: Abstract Line 26: “compared to recipients with Sham monocytes… a systemic effect of..”. Kindly clarify the meaning of Sham monocytes.
- Response 1: Thank you. We modified and clarified the Sham group on lines 21 and 26.
Comment 2, 3: Line 123~124: quantified in the mice's feet, and the ratio between ischemic and non-ischemic was determined to avoid variation of body temperature between the mice[1] Please clarify the statement “to avoid variation of body temperature between the mice.”
- Response 2, 3: Thank you. We modified and clarified why the ratio between the ischemic and non-ischemic foot is important because each mouse's body temperature can influence the absolute value.
- Lines 123-129: “Since body temperature could vary between the mice and influence the absolute measurement of the blood flow in the foot, the ratio between the ischemic and non-ischemic foot should always be considered to avoid misinterpretation of absolute values”
Comment 4: Line 131: "24h trained and Smice, BM-Mono … the ischemic limb (left leg) or the". What does Smice refer to?
- Response 4: It was a typo – Sham, we corrected the text, thanks for that.
Comment 5: Line134: “Miltenyibiotec MACS (cat. no. 130-100-629), .. instructions “Please correct the manufacturer spelling to Miltenyi Biotec
- Response 5: Done, thank you.
Comment 6: Line 135~136: Recipient mice were C57Bl6J, subjected to permanent hindlimb ischemia one day before receiving 1x106 monocytes via tail vein injection.
Please justify the reason for selecting the tail vein injection as the route of administration of monocytes rather than the ischemic site. Furthermore, how is the dose of 1x106 monocytes optimized? Please show the optimization experiment if the authors have performed any.
- Response 6: We published a similar methodology at ATVB 2 years ago and would like to maintain the same methodology since our previous data showed remarkable data (ATVB 2022).
Comment 7: Line 166: Plasma EVs were isolated from 24h trained and Sham groups and sent to Alpha Nano.
Comment 8: Please clarify the process for the isolation of EV or cite the appropriate reference.
- Response 7,8: Thank you. We added information about plasma isolation in the text.
Comment 9: Kindly include the LDPI scale in Figure 1. It is better to include D1 images to observe the improvement in the mice limb.
- Response 9: We included the Images from pre-, post-, and Day 3, but the figure looks a bit busy now. We don’t have a scale for the LDI images, and we have never used one for any of our publications.
Comment 10: Please clarify the number of animals taken for the experiment in Fig 2.
- Response 10: Figure 2 shows that n=3 was used because of the number of dots in each bar. Figs. 2, D, and E are representative pictures.
Comment 11: Line 268: "using electronic microscopy (Figure 2D)...together, suggest that the" Which electronic microscopy was used for this experiment?
- Response 11: The electronic microscopy used for this experiment is described in the methods section 2.8.
Comment 12: Fig 2E : Scale bar is missing in EVs by TEM.
- Response 12: Done, thank you!
Comment 13: Fig 3: Please incorporate Day 0 and Day 3 images for better comparison and to observe the progress.
- Response 13: Done; the figure looks much better, as requested by the reviewer. Thank you
Comment 14: Line 356: “and …model of hindlimb ischemia (our ATVB paper)” The authors are advised to properly reference the paper.
- Response 14: Thank you. We should have caught this mistake earlier but corrected it in the text now.
Comment 15: Line 299~300: (since our previous data showed downregulation of Dhcr24 gene expression and desmosterol accumulation1) we confirmed protein expression of Dhcr24
Line 324~325: Sham BM-Mono (Figure 5A). We repeated the same experimental design using WT BM- Mono treated with SH42, a Dhcr24 inhibitor1.
What does superscript 1 refer to in the above statements?
- Response 15: Thank you. Both mistakes have already been corrected in the manuscript.
Comment 16 Is Figure 3: Left panel graph the quantification of LDPI? It is a good idea to incorporate D0 and D3 images as well.
- Response 16: We believe this is the same question asked on 11)
Comment 17 Fig 3, n=3 seems the sample size is very small.
Does the result from this sample size give reliable results?
- Response 17: The sample size is small, an n=3, but it shows that this data is very powerful because even though we have an n=3, we still observe significant differences between the groups. Unfortunately, we couldn’t increase the sample size since these cells are from a specific knockout colony for Dhcr24 and were shipped from the University of Cincinnati (Dr. Patel’s lab). They shipped us the bone marrow cells, and we isolated the monocytes so we could have a number of monocytes sufficient to inject into 3 recipient mice.
Comment 18Fig 4A: Dhcr expression in both Sham and 24h treated groups are the 4 wells, protein samples from quadruplets, or 4 independent experiments. Please quantify the western blot results.
- Response 18: These are 4 independent experiments, cells isolated from different animals.
Comment 19: Y axis label is missing in Figure 4B.
Figure 4 C: Kindly clarify label on Y axis “ Fold Change vs Sham is the”
- Response 19: Thank you. We labeled the Y axis in Figure 4C and explained the “fold change vs. sham” in the legend.
Comment 20: Figure 5: Please state the fluorescence unit on the Y axis.
Fig 5a: Kindly clarify which staining is shown in the image. Please replace the fluorescence image if possible.
- Response 20:Thank you. We defined the fluorescence unit and clarified the staining in the legend; we also improved the fluorescence images.
Comment 21: The authors are suggested to add the Limitation of this study.
- Response 21: Thank you for this suggestion; a limitation paragraph was added to the discussion
- Lines 418-424 “The limitations of our study include animal models, which may not fully replicate human pathophysiology, potentially limiting the translation of the findings to clinical settings. Additionally, the sample size of this study is small due to the limited number of monocytes that can be extracted from each mouse, which sometimes leads to a pool of few mice for each sample. Lastly, this study focused on a specific gene regulation (Dhcr24) in a specific cell type (monocytes), challenging the idea of using a global inhibition of Dhcr24, an important enzyme in the cholesterol pathway”
Comment 22: It is a good idea to include a flowchart to show the experiments/findings to grasp the audience's attention.
- Resposne 22: We respectfully disagree with the reviewer. We believe that a flow chart with the list of experiments and findings could distract the audience and decrease interest in the manuscript. We would like to keep the manuscript as it was written, improved by most of the reviewers' comments and suggestions.
Reviewer 3 Report
Comments and Suggestions for Authors
1. In this study, authors confirmed that ischemic-trained BM-Mono has an activated desmosterol pathway, that they polarize in anti-inflammatory M2-like macrophages, and that they have a lower adhesion rate to endothelial cells. I think the data provided in the manuscript are fully supported the conclusion.
2. I would like to suggest the authors to add some discussion of the reperfusion effect on the ischemic-trained BM-Mono
Author Response
Comment 1: In this study, authors confirmed that ischemic-trained BM-Mono has an activated desmosterol pathway, that they polarize in anti-inflammatory M2-like macrophages, and that they have a lower adhesion rate to endothelial cells. I think the data provided in the manuscript are fully supported the conclusion.
- Response 1: Thank you!
Comment 2: I would like to suggest the authors to add some discussion of the reperfusioneffect on the ischemic-trained BM-Mono.
- Response 2: Thank you for this comment; we added it to the discussion.
- Lines 137-384 "This study highlights that ischemia training reprograms BM-Mono in a positive way by downregulating their Dhcr24 expression and differentiating them towards a less inflammatory M2-like macrophage, which are essential cells to improve the outcomes in limb ischemia[1]. Our ischemia training consists of 24 hours of ischemia by femoral artery occlusion followed by 48 hours of reperfusion before BM-Mono isolation. Although it is known that reperfusion causes an excessive reactive oxygen species, inducing a pro-inflammatory environment and worsening conditions already compromised by reduced blood flow and oxygen [36], our data demonstrated a positive reprogramming of the BM-Mono. This positive effect might be explained because we assessed the monocytes inside the bone marrow and not the ones in the ischemic site that are subject to a direct effect of the reperfusion or because those BM-Mono were tested 2 days after the blood flow was restored when the hostile environment caused by reperfusion was diminished"